# High *FREM2* Gene and Protein Expression Are Associated with Favorable Prognosis of *IDH*-WT Glioblastomas

**DOI:** 10.3390/cancers11081060

**Published:** 2019-07-27

**Authors:** Ivana Jovčevska, Alja Zottel, Neja Šamec, Jernej Mlakar, Maxim Sorokin, Daniil Nikitin, Anton A. Buzdin, Radovan Komel

**Affiliations:** 1Medical Centre for Molecular Biology, Institute of Biochemistry, Faculty of Medicine, University of Ljubljana, 1000 Ljubljana, Slovenia; 2Institute of Pathology, Faculty of Medicine, University of Ljubljana, 1000 Ljubljana, Slovenia; 3Laboratory of Clinical and Genomic Bioinformatics, I. M. Sechenov First Moscow State Medical University, 119146 Moscow, Russia; 4Omicsway Corp., Walnut, CA 91789, USA; 5Shemyakin-Ovchinnikov Institute of Bioorganic Chemistry, Russian Academy of Sciences, 117997 Moscow, Russia; 6Oncobox Ltd., 121205 Moscow, Russia

**Keywords:** glioblastoma, malignancy, *FREM2*, *SPRY1*, TCGA

## Abstract

World Health Organization grade IV diffuse gliomas, known as glioblastomas, are the most common malignant brain tumors, and they show poor prognosis. Multimodal treatment of surgery followed by radiation and chemotherapy is not sufficient to increase patient survival, which is 12 to 18 months after diagnosis. Despite extensive research, patient life expectancy has not significantly improved over the last decade. Previously, we identified *FREM2* and *SPRY1* as genes with differential expression in glioblastoma cell lines compared to nonmalignant astrocytes. In addition, the *FREM2* and *SPRY1* proteins show specific localization on the surface of glioblastoma cells. In this study, we explored the roles of the *FREM2* and *SPRY1* genes and their proteins in glioblastoma pathology using human tissue samples. We used proteomic, transcriptomic, and bioinformatics approaches to detect changes at different molecular levels. We demonstrate increased FREM2 protein expression levels in glioblastomas compared to reference samples. At the transcriptomic level, both *FREM2* and *SPRY1* show increased expression in tissue samples of different glioma grades compared to nonmalignant brain tissue. To broaden our experimental findings, we analyzed The Cancer Genome Atlas glioblastoma patient datasets. We discovered higher *FREM2* and *SPRY1* gene expression levels in glioblastomas compared to lower grade gliomas and reference samples. In addition, we observed that low *FREM2* expression was associated with progression of *IDH*-mutant low-grade glioma patients. Multivariate analysis showed positive association between *FREM2* and favorable prognosis of *IDH*-wild type glioblastoma. We conclude that *FREM2* has an important role in malignant progression of glioblastoma, and we suggest deeper analysis to determine its involvement in glioblastoma pathology.

## 1. Introduction

Based on their origins, primary brain tumors are broadly categorized as glial, glio-neuronal, embryonic, tumors of the meninges, mesenchymal tumors, tumors of the choroid plexus, tumors of the hematopoietic system, pituitary tumors, and tumors of the sellar region [1,2]. Gliomas are tumors that originate from the supportive glial cells—ependymal cells, astrocytes, and oligodendrocytes, and they are referred to as ependymomas, astrocytomas, and oligodendrogliomas, respectively. The most common type of glial tumors is astrocytomas. According to the World Health Organization (WHO), there are four glioma grades: grade I (e.g., pilocytic astrocytomas); grade II (e.g., diffuse astrocytomas); grade III (e.g., anaplastic astrocytomas); and grade IV (i.e., glioblastomas) [2,3]. The most malignant and aggressive form, glioblastoma, accounts for 60% to 70% of all glioma cases [4]. Even with multimodal clinical management (i.e., maximal safe surgical resection, chemotherapy, radiation), the majority of these patients only survive for 12 to 18 months post diagnosis [5,6,7]. Indeed, only 3% to 5% of glioblastoma patients are still alive >3 years after diagnosis, while survivors to 10 years or longer comprise <1% of all patients [8,9]. High glioblastoma lethality is attributed to late diagnosis as a result of nonspecific symptoms, rapid progression, infiltration into surrounding tissues, intracranial location that complicates surgery, and development of resistance to treatment and common recurrence after initial treatment [10,11,12,13]. In cases of recurrent glioblastomas, the average life expectancy of the patient is reduced to 6 months [14,15].

Genetically, glioblastomas represent a diverse disease, with heterogeneity at both the cellular and molecular levels [16,17]. Variations in cell size and type, cell density, genetics, gene expression profile, morphology, phenotype, and necrosis are observed at the inter-tumor and intra-tumor levels [18,19,20]. The Cancer Genome Atlas (TCGA) project (https://cancergenome.nih.gov) has greatly contributed to the identification of the genetic landscape of glioblastomas [21]. During the course of this project, the core genetic changes that were most commonly observed were alterations in receptor tyrosine kinase (RTK)/rat sarcoma (RAS)/PI3K, p53 and retinoblastoma pathways [22,23]. Based on isocitrate dehydrogenase (IDH) 1 and 2 mutation status, glioblastomas are defined as primary (*IDH*-wild-type (WT)), which arise de novo, and secondary (*IDH*-mutant), which evolve from lower grade gliomas [24]. Further genetic analysis has revealed different subtypes with distinct genetic backgrounds (i.e., proneural, classical, mesenchymal) that can be simultaneously present in a single tumor [1,24,25,26]. Additionally, three histological variants of primary glioblastomas have been reported, as epithelioid glioblastomas, gliosarcomas, and giant-cell glioblastomas [2,27]. However, the 2007 WHO classification recognizes only two formal variants: gliosarcomas and giant-cell glioblastomas, which comprise 2% and 5% of all glioblastomas, respectively [3,28]; thus epithelioid glioblastomas are considered as a provisional entity [29]. This heterogeneous nature of glioblastomas complicates their clinical management. The alkylating agent temozolomide has been part of the established clinical care for patients with glioblastomas since 2005 [22,30,31,32]. In addition, the anti-angiogenic humanized monoclonal antibody bevacizumab has been part of their adjuvant treatment since its approval by the US Food and Drug Administration (FDA) in 2009 [33,34].

In our previous study, we carried out a meta-analysis of the data from publicly available repositories, whereby we selected two genes that show selectivity towards glioblastoma: those for FRAS1-related extracellular matrix protein 2, *FREM2*, and sprouty RTK signaling antagonist 1, *SPRY1* [35]. We showed increased *FREM2* gene expression and FREM2 protein expression levels in glioblastoma cells, compared to nonmalignant astrocytes. Moreover, both FREM2 and SPRY1 showed specific localization to the surface of glioblastoma cells, which was not observed in the case of the reference nonmalignant astrocytes. This surface expression makes FREM2 and SPRY1 suitable for targeting purposes.

To the best of our knowledge, our present study is the first to investigate in depth the involvement of *FREM2* and *SPRY1* in glioblastomas at different molecular levels using human samples, as well as using advanced in-silico approaches to enlarge the sample size. To evaluate potential use of our previous findings for clinical purposes, we performed a pilot study where we examined the expression of *FREM2* and *SPRY1* at the proteomic and transcriptomic levels in different grades of gliomas and nonmalignant brain-tissue samples. *FREM2* showed notable differences in expression at both levels, while *SPRY1* showed differential changes in expression mostly at the transcriptomic level. These data were also confirmed with immunohistochemistry using various glioma tissue samples. Finally, we performed in-silico analysis using TCGA database, which indicated increased *FREM2* and *SPRY1* gene expression in glioblastomas compared to lower grade gliomas and reference samples. In addition, there was an association between decreased *FREM2* gene expression and bad outcome in *IDH*-mutant lower grade gliomas. Finally, these data show that increased *FREM2* gene expression is associated with favorable prognosis for patients with *IDH*-WT glioblastomas. Our data show lower *FREM2* gene expression for patients with *IDH*-WT gliomas whose disease progressed after temozolomide treatment. The results of this study suggest the involvement of the *FREM2* gene and its protein in glioblastoma pathogenesis, and they will serve as the basis for further evaluation of their roles in glioblastoma progression.

## 2. Results

### 2.1. The FREM2 Protein Shows Higher Levels in Glioblastomas Versus Lower Grade Gliomas

To detect changes at protein levels, we performed three different analyses: immunoblotting, ELISA, and immunohistochemistry. The data obtained across these analyses were consistent, and are detailed below.

Representative immunoblots are shown in Figure 1A,B. Quantification of the bands from the immunoblots defined higher FREM2 expression levels in glioblastomas for GBM versus REF (****, *p* < 0.0001) and GBM versus LGG (***, *p* = 0.0009), as shown in Figure 1C. Differences in *FREM2* expression levels were supported also by the ELISA experiment, for GBM versus REF (*p* < 0.0001) and GBM versus LGG (*p* = 0.0032) (Figure 2). Immunohistochemistry additionally confirmed these findings, and the results are presented in Table 1. Quantification of the immunohistochemistry results showed difference in expression of FREM2 in GBM compared to LGG (*p* = 0.0211) (Figure 3). Changes in SPRY1 protein expression levels were observed only in the ELISA experiment for GBM versus REF (*p* = 0.0120) (Figure 2). Neither the immunoblotting (Figure 1B,D) nor the immunohistochemistry (Figure 3A,B) showed significant changes in SPRY1 expression levels among these samples.

### 2.2. The FREM2 and SPRY1 Genes Show Higher Expression in Glioblastomas Versus Lower Grade Gliomas

Using qPCR, we examined the patterns of the gene expression levels of *FREM2* and *SPRY1* in glioma tissue samples of different WHO grades, as well as in nonmalignant brain tissue samples. At the transcriptomic level, both *FREM2* and *SPRY1* showed differences in expression when analyzed as GBM versus LGG (***, *p* = 0.0001; ****, *p* < 0.0001; respectively); i.e., higher gene expression levels in GBM in both cases, as shown in Figure 4. In addition, *FREM2* was expressed at lower levels in LGG compared to REF (**, *p* = 0.0062), while *SPRY1* showed higher expression in GBM compared to REF (*p* = 0.0011) (Figure 4).

### 2.3. FREM2 and SPRY1 Show Lower Expression in IDH-Mutant Low-Grade Gliomas that Progress After First-Line Temozolomide Treatment

Next, to increase the sample numbers used in this analysis and to confirm these experimental findings, we broadened our study to include a bioinformatics analysis of large datasets available from TCGA database. Our in-silico validation took into consideration a large number of cancer types, and showed that median *FREM2* and *SPRY1* expression were comparable among the different cancer types (Figure 5A). When compared among different brain cancer types, both *FREM2* and *SPRY1* showed significantly higher expression in GBM versus LGG (3-fold, 10-fold, respectively), as shown in Figure 5B. *FREM2* and *SPRY1* expression in the reference nonmalignant brain tissue samples (norm.) was significantly lower than in GBM (again, 3-fold, 10-fold, respectively), but not significantly different from LGG (Figure 5B).

Unexpectedly, although with borderline significance (*p* = 0.052), there was lower *FREM2* expression in LGG patients who progressed during first-line temozolomide treatment and had an *IDH* mutation versus patients with *IDH*-mutant stable/responsive disease; whereas there was no difference in *IDH*-WT patients (Figure 6). We observed the same correlation (with significance now reached; *p* = 0.025) for *SPRY1* expression with treatment outcome in all temozolomide-treated *IDH*-mutant positive patients—lower *SPRY1* expression in temozolomide-treated patients who progressed during first-line temozolomide treatment and had *IDH* mutation versus patients with *IDH*-mutant stable/responsive disease (Figure 6C). Additionally, there was no correlation between *SPRY1* gene expression and treatment outcome in the *IDH*-WT patients treated with temozolomide (Figure 6D).

### 2.4. High FREM2 Expression Is Positively Associated with IDH-WT Glioblastoma Patient Survival and Negatively with IDH-WT Low Grade Glioma Patient Survival

To delineate the impact of *FREM2* and *SPRY1* gene expression on the patients’ responses, we performed multivariate survival analysis following Cox proportional-hazards models. Whereas clinical response data were available from TCGA only for low grade glioma patients, overall survival data were available for both low grade glioma and glioblastoma. We took *FREM2* and *SPRY1* gene expression, age, sex, and histological type of brain tumor as covariates (Figure 7). In all cases (glioblastoma *IDH*-WT, low grade glioma *IDH*-WT and mutant), *SPRY1* expression did not significantly impact upon patient overall survival. However, the role of *FREM2* was controversial. *FREM2* was positively associated with survival in glioblastoma *IDH*-WT (Figure 7A) and negatively associated in low grade glioma (Figure 7B) *IDH*-WT (all associations were significant, *p* < 0.05). As expected, age showed a small but significant decrease in the patient overall survival in all cases.

## 3. Discussion

Our current data provide new insight into glioblastoma proteomic and transcriptomic changes. The study was based on previous data [35] where we presented differential expression levels of the *FREM2* and *SPRY1* genes and their proteins in different glioblastoma cells, compared to astrocytes. In the present study we used instead patient samples to confirm this involvement of the *FREM2* gene and its protein in glioblastoma pathology as the malignancy progresses, as shown by the higher *FREM2* expression levels in glioblastoma tissues compared to lower grade gliomas and reference samples. We also observed differences in *FREM2* expression between *IDH*-mutant and *IDH*-WT glioblastomas. Namely, when compared to expression levels in samples from patients whose disease did not progress, in patients who progressed during first-line temozolomide treatment their *FREM2* gene expression levels were halved (*p* = 0.052) in *IDH*-mutant gliomas, while there was no change in gene expression levels in *IDH*-WT glioma samples. The same was observed for *SPRY1*, where gene expression levels in *IDH*-mutated tumors were significantly lower (*p* = 0.025) than in samples from patients whose disease did not progress. Finally here, using multivariate analysis we showed a positive correlation between *FREM2* and survival of patients with *IDH*-WT glioblastomas, and a negative correlation between *FREM2* and survival of patients with *IDH*-WT lower grade gliomas.

### 3.1. FRAS1-Related Extracellular Matrix Protein 2—FREM2

FREM2 is a transmembrane protein that is localized to the cell basement membrane and is associated with cell migration and motility [36,37]. Due to its membrane association, and considering its overexpression is related to tumor grade and progression, FREM2 might be an attractive target for glioblastoma cell targeting.

We observed that FREM2 protein expression levels increased with glioma grade progression. This observation supports the findings of Nagaishi et al. who reported amplification of the *FREM2* gene and overexpression of the FREM2 protein in 64 gliosarcomas [38]. They further investigated different areas of the analyzed gliosarcomas, and found amplification of the *FREM2* gene and overexpression of the FREM2 protein in the mesenchymal areas, and not in the glial tumor areas, of the gliosarcomas. Likewise, Oh et al. reported restricted expression of *FREM2* in the mesenchymal areas of their analyzed fraction of gliosarcomas [28]. Our experimental data also suggest the involvement of *FREM2* in glioblastoma pathogenesis. To confirm our findings on a larger cohort of patients, we performed an expanded in-silico analysis using the data available from TCGA. Indeed, according to this in-silico analysis of TCGA data, *FREM2* expression is lower in tissue samples from patients with low grade *IDH*-mutant gliomas that progressed after temozolomide treatment than in patients with stable or responsive disease. Moreover, multivariate analysis showed that *FREM2* gene expression is positively correlated with patient overall survival in glioblastoma and negatively correlated in *IDH*-WT low grade glioma. These findings are not so unexpected, because some clinically relevant mutations, such as TERT, have provided controversial predictions for glioblastoma due to various confounding factors [39].

The involvement of *FREM2* in glioblastoma progression might also be correlated to the presence of hypermutation, which can commonly occur as a result of chemotherapy treatment [40]. Temozolomide is an oral chemotherapeutic drug, a DNA-alkylating agent, that can be used as first-line treatment for glioblastomas. The therapeutic benefit here is a result of the formation of O^6^-guanine residues that cause mispairing with thymine, and the consequent DNA damage that triggers apoptosis. In addition to this, temozolomide has undesired genotoxic properties that can cause the onset of genetic mutations upon relapse that were not present at diagnosis [31,41,42,43]. The efficacy of temozolomide is also limited by its toxic effects on tissues outside of the central nervous system, as well as the biological limits to the maintenance of a constant tumoricidal concentration, which is the case in most systemic therapies [44]. In addition, variant allele frequency enrichment of *FREM2* in temozolomide-resistant glioblastoma cells has been reported previously [36]. The authors of this study performed whole exome deep sequencing of in-vitro temozolomide-treated residual cell cultures, and reported a 76% variant allele frequency enrichment of *FREM2* in their aggressive glioblastoma-derived neurospheres. It is now a challenge for the research community to determine whether the *FREM2* changes detected are a cause for or a consequence of glioblastoma formation.

### 3.2. Sprouty RTK Signaling Antagonist 1—SPRY1

SPRY1 has already been correlated to different cancers, but the published data are conflicting. On the one hand, *SPRY1* is considered a candidate tumor-suppressor gene due to its down-regulation in breast, prostate, and liver cancers [45]. It has been suggested that increased SPRY1 expression leads to inhibition of tumor growth in human breast cancer cells [46]. Moreover, Liu et al. also showed that overexpression of SPRY1 in different cell lines inhibits their proliferation [45]. On the other hand, *SPRY1* represents an oncogene in triple negative breast cancer cell lines [47], and was also reported to be overexpressed in hepatocellular carcinoma [46]. This implies that its role might be cancer specific. The role of SPRY1 in glioblastoma cell lines was also examined and it was shown that *SPRY1* knock-down reduced expression of mesenchymal markers and impaired invasiveness of U251 cells [47]. At the transcriptomic level, our data are in agreement with the findings of Liu et al. [45], whereas at the proteomic levels we did not observe the same effects as those reported by others with the exception of the results from our ELISA experiment. The SPRY1 role in cancer development is believed to be through acting as a negative feedback inhibitor of RTK signaling (pathways that are commonly altered in many cancers), and as an inductor of cellular senescence [45,48,49]. In addition, *SPRY1* is one of the predicted targets of miR-21, which is over-expressed in glioblastoma-initiating cells [50]. The authors of the latter study showed that miR-21 overexpression induces decreased SPRY1 protein expression. Taking in consideration all of these reports and our findings, the mechanism by which *SPRY1* acts in a protective and/or oncogenic role in different cancers is yet to be defined.

## 4. Materials and Methods

### 4.1. Ethics Statement

This study was approved by the National Medical Ethics Committee of Republic of Slovenia, approval numbers 92/06/12, 89/04/13, 95/09/15, 0120-196/2017/7 and 0120-190/2018/4. Reference samples were obtained during routine autopsies following national legal regulations of the Republic of Slovenia. Patients signed written informed consent prior to their surgery. All of the samples used in this study are anonymous.

### 4.2. Tissue Samples

WHO grade II and III gliomas (i.e., lower grade gliomas) were obtained from 13 patients (nine males, four females) aged 25 to 53 years. Tissue samples of WHO grade IV gliomas (i.e., glioblastoma) were obtained from 12 patients (eight males, four females) aged 41 to 81 years. The patient clinicopathological features are presented in Table 2. The extended data are available as Appendix A. Reference post mortem brain tissue samples from hippocampus, subventricular and periventricular zones of 10 patients were obtained during autopsies (Table 3). After dissection, all of the glioma and reference samples were sealed in sterile containers, labelled, and snap frozen. All of these tissue samples were kept at −80 °C until used for protein isolation and RNA extraction.

### 4.3. Proteomic Analysis

#### 4.3.1. Protein Isolation

Proteins were extracted from 12 glioblastoma samples, 10 grade II glioma samples (proteins could not be isolated from one sample as there was insufficient tissue), two grade III glioma samples and six reference samples originating from the hippocampus (two samples), periventricular zone (two samples), and subventricular zone (two samples) of post-mortem brain tissue samples, using ProteoExtract Transmembrane Protein Extraction kits (71772-3; Novagen, Madison, WI, USA), following the manufacturer instructions. Protein concentrations were determined using Pierce BCA Protein Assay kits (23227; Thermo Scientific, Waltham, MA, USA).

#### 4.3.2. Immunoblotting

Approximately 20 µg of each protein extract was used for the immunoblotting. To analyze all of the samples, we loaded four glioblastomas, four grade II/III gliomas, and six reference samples per gel. Protein extracts were separated using 4% to 12% NuPAGE Bis-Tris Mini Gels (NP0321BOX; Invitrogen, Carlsbad, CA, USA), and transferred to Immobillion-P PVDF transfer membrane (IPVH00010; Merck-Millipore, Burlington, MA, USA). PageRuler Plus Prestained Protein Ladder (26619; Thermo Scientific) was used as a molecular marker. Whole proteins were reversibly colored with Ponceau S. For antigen detection, residual protein binding sites were blocked with 5% phosphate-buffered saline (PBS)-milk, with shaking at 60 rpm for 1 h at 4 °C. Mouse monoclonal anti-SPRY1 antibody (WH0010252M1; Sigma Aldrich, St. Louis, MO, USA) and rabbit polyclonal anti-FREM2 antibody (SAB3500517; Sigma Aldrich) were used in combination with FREM2 blocking peptide (SBP3500517; Sigma Aldrich). The mouse monoclonal anti-GAPDH antibody (G8795, Sigma Aldrich) was used as the loading control. Incubations with primary antibodies for antigen detection were overnight at 4 °C while shaking at 60 rpm. Incubations with antibodies for the loading controls and secondary anti-mouse (A4416; Sigma Aldrich) and anti-rabbit (A0545; Sigma Aldrich) IgG whole molecule horseradish peroxidase antibodies produced in goat were for 1 h at 4 °C, while shaking at 60 rpm. Bands were revealed with SuperSignal West Pico PLUS Chemiluminescent Substrate (34580; Thermo Scientific), visualized using a CCD camera (FujiFilm LAS-4000; FujiFilm, Tokyo, Japan), and analyzed with the Multi Gauge version 3.2 software (FujiFilm, Tokyo, Japan). Relative band intensities were calculated as $\frac{AU\left(antigen\right)}{AU\left(AVERAGEreference\right)}$. GAPDH was included to show equal protein loading, and therefore it was not quantified.

#### 4.3.3. Enzyme-Linked Immunosorbent Assay

Isolated proteins from the tissue samples were coated overnight at 4 °C onto NUNC Maxisorp enzyme-linked immunosorbent assay (ELISA) plates at 2 µg/mL and 100 µL/well. The coating buffer was 0.1 M NaHCO_3_, pH 9. The following day, the wells were washed with PBS supplemented with 0.01% Tween, and the residual binding sites were blocked with 5% PBS-milk. The wells were washed again, and 100 µL primary antibodies (1:2000; same as those used during the immunoblotting) were put into each well. After 1 h incubation at room temperature, the wells were washed again and secondary anti-mouse (A3562, Sigma Aldrich) and anti-rabbit (A3687, Sigma Aldrich) IgG whole molecule alkaline phosphatase antibodies produced in goat were prepared at 1:2000 dilution in 1% PBS-milk, and were applied (100 µL/well). After 1 h incubation at room temperature, the wells were washed again, and 100 µL/well alkaline phosphatase substrate (P4744; Sigma Aldrich) was added. Signals were measured using a microplate reader (Synergy H4 Hybrid; BioTek, Winooski, VT, USA) at 405 nm.

#### 4.3.4. Immunohistochemistry

Hematoxylin and eosin (H&E) staining and immunohistochemistry were performed on formalin-fixed paraffin-embedded samples from four oligodendrogliomas, four diffuse astrocytomas, two anaplastic astrocytomas, eight glioblastomas, one epithelioid glioblastoma, one recurrent glioblastoma, and one secondary gliosarcoma. Brain glial cells and neuropil were used as the negative control for FREM2. Liver bile duct and brain glial cells and neuropil were used as the negative control for SPRY1. Monoclonal antibodies (same as those used for immunoblotting and ELISA) were used at 1:1000 dilution. An automatic staining machine (Ventana Discovery, Roche, Basel, Switzerland) was used, and the samples were thermally processed with CC1 buffer (951-124; VENTANA). Detection was carried out using iVIEW Dab Ventana.

The samples were analyzed by a pathologist and graded from “+” as the lowest reaction intensity, to “+++” as the strongest reaction intensity. To perform statistical analysis, the percentages of positive cells were calculated as: 0−25%, 1; 25−50%, 2; 50−75%, 3; and 75−100%, 4. The samples were grouped as Grade IV, which consisted of primary and recurrent glioblastomas, secondary gliosarcoma, and epithelioid glioblastoma; and Grade II, which consisted of diffuse astrocytoma and oligodendroglioma. Anaplastic astrocytomas were not used for statistics due to the small number of samples.

### 4.4. Transcriptomic Analysis

#### 4.4.1. RNA Extraction

RNA was extracted from all of the available samples using TRI reagent (T9424; Sigma Aldrich) as described by the manufacturer. RNA concentrations were measured using a NanoDrop ND-1000 (NanoDrop Technologies, Wilmington, DE, USA) and purity was determined using A_260_/A_280_ and A_260_/A_230_ ratios. RNA integrity was determined using a bioanalyzer (2100; Agilent Technologies, Santa Clara, CA, USA).

#### 4.4.2. Quantitative Real-Time Polymerase Chain Reaction

Quantitative real-time polymerase chain reaction (qPCR) was performed as previously described [51]. Briefly, 2 µg of each RNA sample was treated with recombinant RNAse-free DNAse I (04716728001; Roche, Basel, Switzerland) for 15 min at 30 °C, 10 min at 75 °C, and then transcribed using Transcriptor Universal cDNA Master (05893151001; Roche, Basel, Switzerland), for 5 min at 25 °C, 10 min at 55 °C, and 5 min at 85 °C. qPCR was performed using Roche LightCycler 480 platform (Roche, Basel, Switzerland). The 5 µL reaction volume consisted of 0.75 µL cDNA, 2.5 µL 2× LightCycler 480 SYBR Green I Master (Roche), 0.3 µL each 2.5 mM primer, and 1.15 µL distilled H_2_O. The following thermal cycling was used: pre-incubation 10 s at 95 °C; cycling 20 s at 60 °C, and 20 s at 72 °C for 45 cycles; melting curve 5 s at 95 °C, and 1 min at 65 °C; continuous at 97 °C, and cooling for 30 s at 4 °C. Five candidate normalization genes *TBP*, *HPRT1*, *RPL13A*, *GAPDH*, and *CYC1* were selected from the literature [52,53,54,55]. In our sample cohort, *RPL13A* and *CYC1* showed the most stable expression patterns and were chosen as normalization genes using the NormFinder algorithm [52]. Primer sequences obtained from the PrimerBank PCR primer database for quantitative gene expression analysis (https://pga.mgh.harvard.edu/primerbank/) were used in our previous publications [35,51] and are given in Table 4. Relative quantification was performed as described previously [56].

### 4.5. Statistical Analysis

Statistical analysis was performed using GraphPad Prism 6 (GraphPad Software Inc., La Jolla, CA, USA). For samples that followed a Gaussian distribution, the analysis was performed using One-way ANOVA and Holm-Sidak’s corrections for multiple comparisons. In cases where two groups were analyzed, samples were tested with unpaired two-tailed Student’s *t*-tests with Welch’s correction. For samples not following a Gaussian distribution, Kruskal–Wallis tests with Dunn’s corrections were used. In cases where two groups were tested Mann–Whitney and Kolmogorov–Smirnov tests were used. In all cases, *p* ≤ 0.05 was considered statistically significant (*, *p* ≤ 0.05; **, *p* < 0.01; ***, *p* < 0.001; ****, *p* < 0.0001).

### 4.6. Bioinformatic Analysis

#### In-Silico Analysis Using the Cancer Genome Atlas Datasets

The Cancer Genome Atlas RNA-seq data were retrieved via the command line tool GDC client (https://gdc.cancer.gov/access-data/gdc-data-transfer-tool). The patient clinical information was taken from Ceccarelli et al. [57]. Raw gene counts were merged together for all samples, and normalized via the DESeq procedure. Information about the investigated TCGA cancer samples is given in Table 5. In the analysis, five reference samples (normal), 528 lower grade glioma (LGG), and 167 glioblastoma (GBM) samples were included. Additionally, 88 low grade glioma patients treated with temozolomide (12 cases of progressive disease, 76 cases of stable disease—partial or complete responses) were included in the analysis. These patients were studied in two groups according to their *IDH* mutation status (an important prognostic biomarker): 17 WT patients (eight with progression, nine with stabilization or response), and 71 WT patients (four with progression, 67 with stabilization or response). Only first-line therapy from the time of sample collection was considered. Analysis of *FREM2* and *SPRY1* gene expression with respect to patients’ clinical response status was performed only for low grade glioma patients, because clinical response data are provided only for low grade glioma patients in TCGA repository. Multivariate survival analysis was performed with data on 53 glioblastoma and 98 low grade glioma patients, and according to Cox proportional-hazards model (http://www.sthda.com/english/wiki/cox-proportional-hazards-model). All images were built using R ggplot2 package.

## 5. Conclusions

Glioblastoma is a complex disease that imposes a great burden on both the patients and medical society. As a result of its great heterogeneity, it is difficult to find any universally successful treatment for this incurable disease. This study is part of a wide-ranging effort to search for relevant biomarkers that might improve the state of this field. As such, for the first time, we deeply examined the relationships between both the *FREM2* and *SPRY1* genes and their proteins in glioblastoma. However, data about the roles of these two proteins in glioblastoma remain scarce.

We validated the *FREM2* and *SPRY1* genes and their proteins in glioma tissue samples from different grades with respect to different human cancers and reference samples. A possible role for the *SPRY1* gene in the pathology of glioblastoma has been addressed in multiple cases, as well as in our study, but unfortunately without any clear answers. On the other hand, our findings clearly show differential expression of the FREM2 protein in glioblastoma tissue samples, which might be relevant for further use in clinical practice. In addition, it has been suggested that *FREM2* has a role in glioblastoma pathology and is associated with favorable patient prognosis. Due to the observed changes at both the proteomic and transcriptomic levels, we therefore suggest further exploration of *FREM2* in both primary and recurrent glioblastomas, to evaluate its role in disease progression. We also found a correlation between the *SPRY1* gene expression levels and patients with *IDH*-WT gliomas that were treated with temozolomide. However, due to the limited sample numbers in our study, we recommend moving to larger scale confirmation studies. Finally, for diagnostic purposes, analysis of these expression levels in human biological liquids (e.g., blood, cerebrospinal fluid) should also be considered.

## Figures and Tables

**Figure 1 cancers-11-01060-f001:**
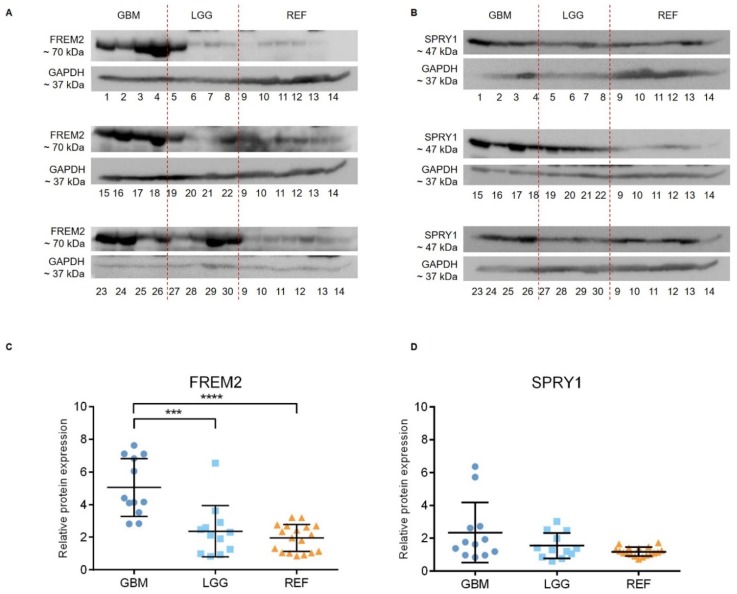
Analysis of FREM2 (**A**,**C**) and SPRY1 (**B**,**D**) protein expression levels using immunoblotting. Glioblastoma samples (GBM: 1, 2, 3, 4, 15, 16, 17, 18, 23, 24, 25, 26), lower grade glioma samples (LGG: 5, 6, 7, 8, 19, 20, 21, 22, 27, 28, 29, 30) and reference brain samples (REF: 9, 10, hippocampus; 11, 12, periventricular zone; 13, 14, subventricular zone) are shown. (**A**,**B**) Representative immunoblots of the proteins of interest. (**C**,**D**) Quantification of the immunoblotting. Data are means ± standard deviation, and show significantly higher protein expression levels of FREM2 in GBM versus REF (****, *p* < 0.0001) and GBM versus LGG (***, *p* = 0.0009). Significant differences were not observed for SPRY1 expression levels among these analyzed samples.

**Figure 2 cancers-11-01060-f002:**
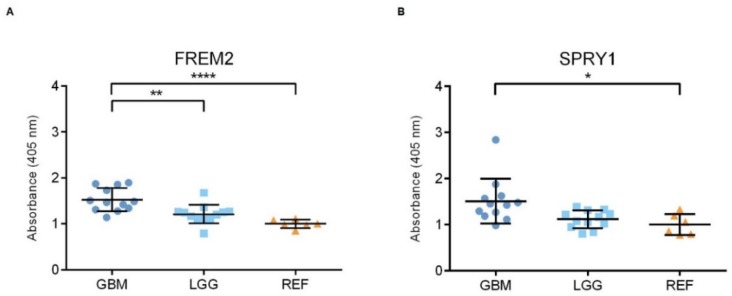
Analysis of FREM2 (**A**) and SPRY1 (**B**) protein expression levels using ELISA. Quantification of the ELISA. GBM, glioblastoma; LGG, lower grade glioma; REF, reference brain samples. Data are means ± standard deviation, and show significantly higher ELISA signals for FREM2 in GBM versus REF (****, *p* < 0.0001) and GBM versus LGG (**, *p* = 0.0032). For SPRY1, there were significantly higher ELISA signals for GBM versus REF (*, *p* = 0.0120).

**Figure 3 cancers-11-01060-f003:**
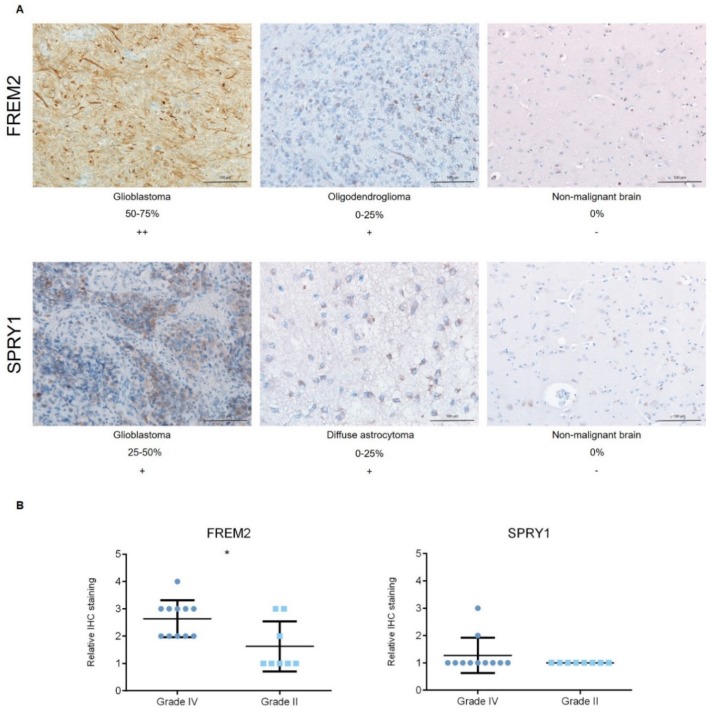
Analysis of FREM2 and SPRY1 protein expression levels using immunohistochemistry. (**A**) Representative immunohistochemistry samples. Magnification: glioblastoma, oligodendroglioma, reference (nonmalignant brain), 200×; (**B**) Quantification of the immunohistochemistry, with proportions of positive cells defined as: 0−25%, 1; 25−50%, 2; 50−75%, 3; and 75−100%, 4. Data are means ± standard deviation. Grade IV, primary and recurrent glioblastomas, secondary gliosarcomas and epithelioid gliomas; Grade II, diffuse astrocytomas and oligodendrogliomas. Significantly higher protein expression levels of FREM2 were shown for Grade IV versus Grade II (*, *p* = 0.0211). Differences were not observed for SPRY1 expression levels among these analyzed samples.

**Figure 4 cancers-11-01060-f004:**
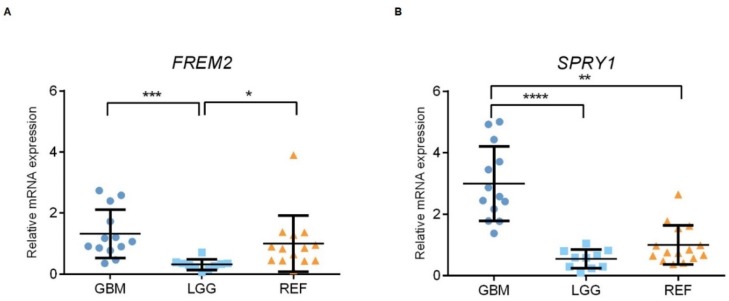
Analysis of *FREM2* (**A**) and *SPRY1* (**B**) mRNA levels using qPCR. GBM, glioblastoma; LGG, lower grade glioma; REF, reference brain samples. Data are means ± standard deviation, whereby both *FREM2* and *SPRY1* show significantly higher expression in GBM versus LGG (***, *p* = 0.0001; ****, *p* < 0.0001; respectively). *FREM2* was expressed at significantly lower levels in LGG versus REF (*, *p* = 0.0137), while *SPRY1* showed significantly increased expression in GBM versus REF (**, *p* = 0.0010).

**Figure 5 cancers-11-01060-f005:**
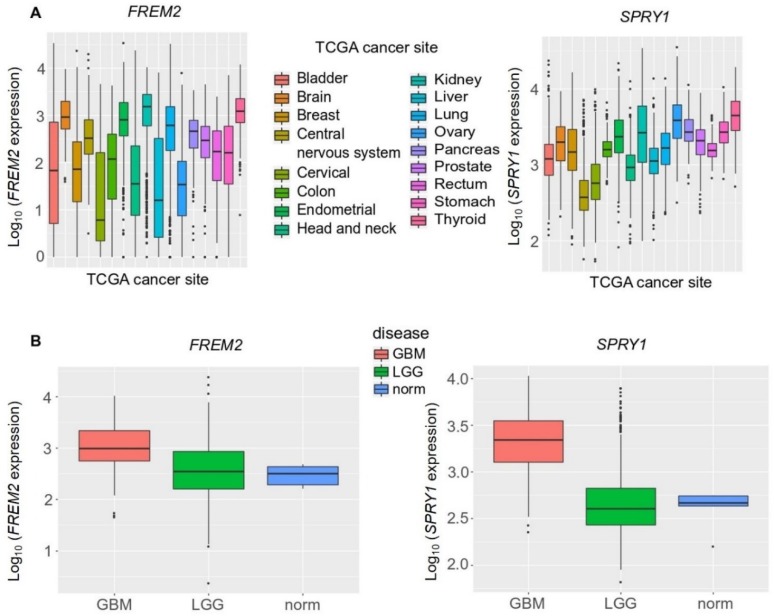
Analysis of *FREM2* and *SPRY1* gene expression in different TCGA (The Cancer Genome Atlas) tumor samples using TCGA data, according to TCGA sites of cancers (**A**) and TCGA brain tumor samples (**B**). GBM, glioblastoma; LGG, lower grade gliomas; Norm, reference nonmalignant brain samples. *FREM2* was expressed at significantly higher levels in GBM versus LGG (*p* < 2.2 × 10^−16^) and GBM versus normal (*p* = 0.003), but without significance in LGG versus normal (*p* = 0.689). Similarly, for *SPRY1*, in GBM versus LGG (*p* < 2.2 × 10^−16^) and GBM versus normal (*p* = 0.0004), and without significance in LGG versus normal (*p* < 0.931).

**Figure 6 cancers-11-01060-f006:**
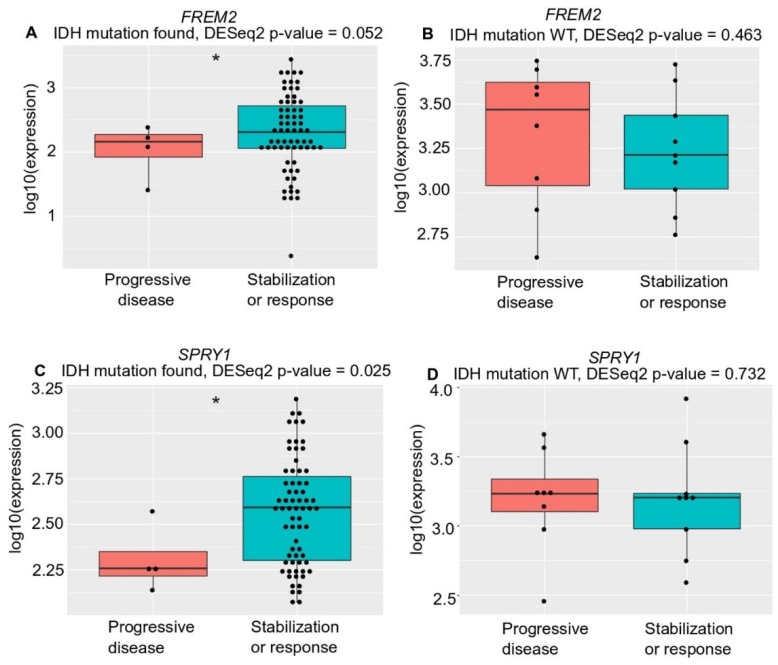
Analysis of *FREM2* (**A**,**B**) and *SPRY1* (**C**,**D**) for TGCA gene expression and temozolomide treatment responses in low grade glioma samples of different *IDH* mutation states, according to progressive disease and stabilization or response. (**A**) *FREM2* expression in *IDH* mutation positive low-grade glioma samples. *FREM2* gene expression levels were significantly lower in *IDH*-mutant gliomas that progressed after temozolomide treatment (*p* = 0.052); (**B**) *FREM2* expression in *IDH* mutation negative (*IDH*-wild type [WT]) low grade glioma samples. No significant differences in gene expression were seen; (**C**) *SPRY1* expression in *IDH* mutation positive low-grade glioma samples. *SPRY1* gene expression levels were significantly lower in *IDH*-mutant positive gliomas that progressed after temozolomide treatment (*p* = 0.025); (**D**) *SPRY1* expression in *IDH* mutation negative (*IDH*-WT) low grade glioma samples. No significant differences in gene expression were seen.

**Figure 7 cancers-11-01060-f007:**
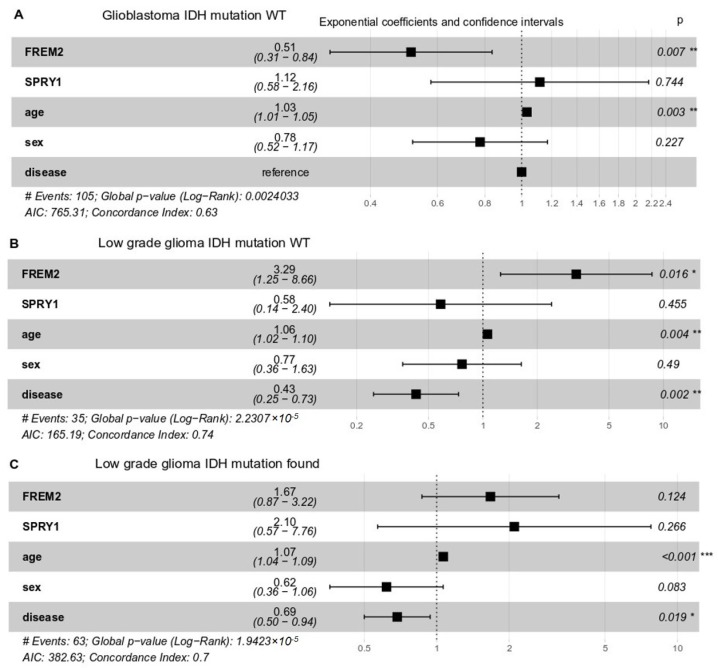
Multivariate survival analysis of overall survival probabilities with respect to *FREM2* and *SPRY1* gene expression, age, sex, and histological type (disease) in TCGA patients with different *IDH* mutation status. Exponential coefficients: according to Cox proportional-hazards models, these are scores that indicate possible impact of a given covariate (e.g., *FREM2* expression, age, sex) on overall survival. An exponential coefficient >1.0 indicates that the given covariate probably decreases patient survival, whereas coefficients <1.0 indicates increased patient overall survival when this covariate in high. Confidence intervals: 95% for exponential coefficients. Generally, if a confidence interval does not contain the point 1.0, the value of the exponential coefficient is considered significant. *P*, Cox proportional-hazards *p*-value for given covariate. (**A**) Glioblastoma, *IDH*-WT, 105 patients; (**B**) Low grade glioma, *IDH*-WT, 35 patients; (**C**) Low grade glioma, *IDH* mutation positive, 63 patients.

**Table 1 cancers-11-01060-t001:** Immunohistochemistry analyses. Pathological examinations were initially graded from “+” as the lowest reaction intensity, to “+++” as the highest reaction intensity. The proportions of positive cells were estimated by a pathologist (0–25%, 25–50%, 50–75%, 75–100%).

Tumor Type	Reaction Intensity		Positive Cells (%)	
	*FREM2*	*SPRY1*	*FREM2*	*SPRY1*
Oligodendroglioma	+	+	0–25	0–25
Diffuse astrocytoma	+	+	0–25	0–25
Primary glioblastoma	++	+	25–50	0–25
Glioblastoma	++	++	75–100	25–50
Oligodendroglioma	+	+	0–25	0–25
Secondary gliosarcoma	++	+	50–75	0–25
Primary glioblastoma	++	+	25–50	0–25
Primary glioblastoma	++	+	25–50	0–25
Glioblastoma	++	+	50–75	0–25
Oligodendroglioma	+	+	0–25	0–25
Epitheloid glioblastoma	++	++	50–75	50–75
Recurrent glioblastoma	++	+	25–50	0–25
Anaplastic astrocytoma	++	+	0–25	0–25
Glioblastoma	++	+	25–50	0–25
Glioblastoma	++	+	50–75	0–25
Glioblastoma	+++	+	50–75	0–25
Diffuse astrocytoma	++	+	50–75	0–25
Anaplastic astrocytoma	++	+	25–50	0–25
Diffuse astrocytoma	++	+	25–50	0–25
Oligodendroglioma	++	+	0–25	0–25
Diffuse astrocytoma	+	+	50–75	0–25

**Table 2 cancers-11-01060-t002:** Clinicopathological features of the glioma patients.

Feature	Detail	Glioma Grade (WHO)		
		II	III	IV
Number of samples	(N)	11	2	12
Gender (*n*)	Female (*n*)	4		4
	Male (*n*)	7	2	8
Age range	(years)	25–53	29–34	41–81
Karnofsky performance scale	(%)	60–100	70–90	40–100
Overall survival	Median (months)	21		15
Patients still alive	(*n*)	8	2	3
Diagnosis (*n*)	Oligodendroglioma	2		
	Diffuse astrocytoma	9		
	Anaplastic astrocytoma		2	
	Glioblastoma			10
	Giant cell glioblastoma			1
	Gliosarcoma			1
Anatomical location (*n*)	Parietal lobe	1 (right)		1 (right)
	Frontal lobe	4 (right) 2 (left)	1 (right)	1 (right) 3 (left)
	Insular cortex	2 (right)	1 (left)	
	Temporal lobe	2 (right)		2 (right) 3 (left)
	Occipital lobe			1 (right)
	Parietal occipital lobe			1 (right)
1p/19q codeletion (*n*)	Positive	2		
	Negative	5		1
	19q deletion	1		
	N/A	3	2	11
*IDH* R132H status (*n*)	Wild-type	2		12
	Mutated	8	2	
	N/A	1		
*ATRX* (*n*)	Loss	6	2	1
	No loss	3		9
	Inconclusive	1		
	N/A	1		2
*TP53* (*n*)	Wild-type	8		8
	Mutated	1	2	4
	Inconclusive	1		
	N/A	1		

WHO, World Health Organization; *IDH*, isocitrate dehydrogenase gene; *ATRX* alpha-thalassemia/mental retardation syndrome gene, X-linked; *TP53*, tumor protein p53 gene, N/A, test not needed or not performed.

**Table 3 cancers-11-01060-t003:** Information about the reference post-mortem brain samples.

Patient Number	Anatomical Location	NO of Samples
1	Hippocampus	1
	Periventricular zone	1
2	Subventricular zone	1
	Hippocampus	1
3	Brain	3
4	Hippocampus	2
	Subventricular zone	1
5	Hippocampus	1
	Periventricular zone	1
6	Periventricular zone	1
	Hippocampus	1
7	Brain	3
8	Subventricular zone	1
9	Periventricular zone	1
	Hippocampus	1
10	Hippocampus	1
	Periventricular zone	1

**Table 4 cancers-11-01060-t004:** qPCR primers for reference genes and genes of interest.

Gene	Primer Pair (5′ → 3′)
*RPL13A*	F: CCT GGA GGA GAA GAG GAA AGA GA R: TTG AGG ACC TCT GTG TAT TTG TCA A
*CYC1*	F: GAG GTG GAG GTT CAA GAC GG R: TAG CTC GCA CGA TGT AGC TG
*FREM2*	F: TGA GCC AAC TGT GTT TAT TC R: GTA TAA CAG ACC ACC ATC AAC
*SPRY1*	F: CTT TGC ATT AGG ATT TCA GAT G R: GGA TCA CAA CTA ACG AAC TG

F—forward, R—reverse.

**Table 5 cancers-11-01060-t005:** Overview of the investigated TCGA samples.

Cancer Site	No of Samples
Rectum	172
Pancreas	182
Cervical	309
Ovary	375
Stomach	407
Liver	424
Bladder	430
Central nervous system	676
Head and neck	546
Prostate	551
Thyroid	568
Endometrium	570
Kidney	1017
Lung	1135
Breast	1216

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
