# Peer review of "High FREM2 Gene and Protein Expression Are Associated with Favorable Prognosis of IDH-WT Glioblastomas"

_cancers, 2019, doi:10.3390/cancers11081060_

Round 1
Reviewer 1 Report
The manuscript has improved a lot. However, there are still a few criticisms which should be addressed:
Fig. 3A: Pictures of non-malignant brains do not contain a proper counterstaining of nuclei by hemataoxilin. Scale bar to inform about the magnification is missing. All pictures should have the same magnification.
Firg. 3B: This graphical presentation does not fit. It should be a bar plot and every bar should represent a staining score for the different WHO grades.
Whenever LGG TCGA data are used, please be precise that these are not low-grade (WHO°II) but lower-grade (WHO°II +III) tumors.
Table 2: A macroscopic evaluation of tumor resection is not possible. This can onlybe done by intra- or early postoperative MRI. If this has not been done please remove this part of the table.
Reviewer 2 Report
A good finding with nicely written manuscript
Reviewer 3 Report
Authors answered comments.
